# High Voltage Redox-Meditated Flow Batteries with Prussian Blue Solid Booster

**John Ostrander, Reza Younesi and Ronnie Mogensen ***

Department of Chemistry–Ångström Laboratory, Uppsala University, Box 538, 75121 Uppsala, Sweden; john.ostrander@kemi.uu.se (J.O.); reza.younesi@kemi.uu.se (R.Y.)

**\*** Correspondence: ronnie.mogensen@kemi.uu.se

**Abstract:** This work presents Prussian blue solid boosters for use in high voltage redox-mediated flow batteries (RMFB) based on non-aqueous electrolytes. The system consisted of sodium iodide as a redox mediator in an acetonitrile catholyte containing solid Prussian blue powder. The combination enabled the solid booster utilization in the proposed systems to reach as high as 66 mAh g$^{-1}$ for hydrated Prussian blue and 110 mAh g$^{-1}$ for anhydrous rhombohedral Prussian blue in cells with an average potential of about 3 V (vs. Na$^+$/Na). Though the boosted system suffers from capacity fading, it opens up possibilities to develop non-aqueous RMFB with low-cost materials.

**Keywords:** flow battery; Prussian blue; solid booster; redox-mediated; non-aqueous

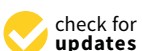



## 1. Introduction

Flow batteries are one of the most promising options to realize medium- to long-term energy storage due to the excellent scalability of these systems combined with the very high safety afforded by separate storage of the anolyte and the catholyte. Currently, the vanadium-based systems dominate the market for flow batteries, and while they have many merits such as high safety, robust chemistry, and long lifetimes, they also have some undesirable properties [1,2]. First and foremost is the use of expensive vanadium and Nafion membranes, which makes huge battery installations quite costly. Secondly, they have very limited energy densities both in regard to weight and volume, and while this is not a huge issue for stationary applications such as peak shaving, load leveling, or backup power where $/kWh is the prime metric, it does rule out the majority of mobile applications where space or weight is limited [3]. Thirdly, while the performance of Nafion membranes is rather good in terms of ionic conductivity, the membranes do suffer from a lack of ion selectivity, which leads to mixing of active molecules from catholyte and anolyte and net transfer of solvent [4].

There are quite mature aqueous chemistries based on iron and zinc that go a long way to alleviate the most pressing concerns of the vanadium RFBs, although some issues such as relatively low voltage and crosstalk through the membranes remain [5,6]. The main reason for the poor energy density is the low voltage that can be realized in aqueous systems and the limitation imposed by solubility of the active materials.

To improve the aforementioned issues, researchers have turned to a relatively new concept where solid boosters are used in redox flow cells using organic or aqueous electrolytes [7–9]. Solid boosters are redox-active solids, i.e., regular anode and cathode materials that enable much higher volumetric energy densities than what is possible even with high concentrations of dissolved compounds. The solid boosters can either be pumped along with the liquid in what would be a slurry type battery or remain stationary in the storage tanks to regenerate a dissolved redox mediator [10].

In theory, slurry type systems would be able to achieve higher energy efficiencies since there is no loss in booster–mediator charge transfer, however, in practice, pumping

slurry is energy demanding. Furthermore, the pumping of suspended solids often leads to clogging and abrasion of pumps, pipes, and membranes.

In the case of redox-mediated systems, the circulated fluid contains dissolved redox-active species that can shuttle electrons while the solids remain in storage tanks. One of the most enticing prospects of redox mediators is that it removes the need for electron percolation in the electrode. Instead of connecting the active material by conductive additives, the electron transport occurs by liquid contact of the dissolved redox mediator and solid booster. This allows for uncoated materials with low electrical conductivity such as phosphorus and Prussian blue to be used in bulk without any costly modifications. Furthermore, the volumetric expansion of materials such as tin or silicon is much easier to handle since particle isolation is not much of an issue.

For the redox-mediated systems there are two main sources of energy losses. The first one occurs when the redox potential of the redox mediator and solid booster is not perfectly matched. The second one is the energy required to pump liquid containing redox mediator through the solid particles in a system that is basically a fluidized bed reactor.

Even though the use of solid booster further complicates the flow battery system, it does enable increased energy density while lowering the volumes of catholyte and anolyte. At the same time, since there are many more redox mediators and solid cathode materials, it is a more flexible system than traditional flow batteries that rely on suitable dissolved compounds.

The use of redox mediators enables the separation of anode and cathode materials to any distance that is feasible to pump. This ability means that this type of cell can be extremely safe with only small amounts of reactive materials in close proximity. The modularity of the design also means that RMFB modules can be serviced by changing solvents and booster materials. Furthermore, expanding the reservoirs to gain energy storage or the cell stack for extra power capability is also possible. An additional benefit of the modularity is that disassembly and recycling become trivial as compared to the mix obtained when shredding lithium-ion batteries.

Since flow cells mostly target large scale energy storage systems, it is important that abundant and cheap materials are used. To this end, iron stands out as one of the best elements for green energy storage since it is abundant, cheap, and provides a respectable voltage [11]. Thus, many of the demonstrations of solid boosters for lithium-based systems have used lithium iron phosphate (LFP) as the booster [12–14]. For sodium-based flow cells, this chemistry is not as favorable due to an electrochemically inactive and thermodynamically stable phase of sodium iron phosphate [15], and in this work the very environmentally benign compound Prussian blue $Na_2Fe_2(CN)_6 \cdot xH_2O$ is used instead.

Prussian blue and its analogs have previously been used in flow batteries [16–18] and provide an average discharge potential of ca 3 V when used in laminated cells as compared to the 3.4 V for LFP while both provide between 150–170 mAh $g^{-1}$ [19–21]. This means that the specific energy calculated using the theoretical capacity and 3 V average discharge potential is close to 500 Wh $kg^{-1}$. The crystal density of anhydrous Prussian blue is quite low for a solid material (ca 1.8 g $cm^{-3}$), but this still translates into a theoretical energy density of more than 900 Wh $L^{-1}$. There are, as always, several practical considerations that prevent the theoretical value from being achieved, but given that vanadium-based systems currently achieve below 100 Wh $L^{-1}$, it is a tempting endeavor to develop redox-flow batteries boosted by Prussian blue.

The targeted cell designs rely on sodium iodide redox mediator in solution that has been previously shown to be able to sodiate the solid Prussian blue [21]. The choice of sodium iodide can be motivated by properties such as a high solubility, reasonable price, and high abundance, although in this work it was chosen because of the two-step reaction from NaI to $NaI_3$ to $I_2$ that gives one mediator two separate redox potentials. Choosing redox mediators often means balancing energy efficiency vs. solid booster utilization. While a single potential redox mediator perfectly matched to a flat voltage plateau solid booster could reach as high as 80% utilization combined with good energy efficiency [22], it

would not be optimal for this system. The sodium iodide allows for high booster utilization and is expected to handle the potential difference between high and low spin iron in rhombohedral Prussian blue, since the use of $I_2$ as an oxidizer and NaI as a reducing agent is equivalent to using two different redox mediators. This approach sacrifices efficiency for simplicity and should be improved upon for any commercial system, especially since iodine and its salts are known for corrosion issues.

Using redox mediators changes the basic rules of the cell design as compared to laminated cells. This change is accompanied with very tempting benefits and very challenging drawbacks that are explored.

Despite the complex electrochemistry, one of the toughest challenges for RMFB research is the engineering. Even lab cells are complex systems with many different components that rely on gas-tight seals and moving parts in combination with brittle membranes (see Figure 1). This in turn means that many factors can influence the experiments and lead to erroneous conclusions.

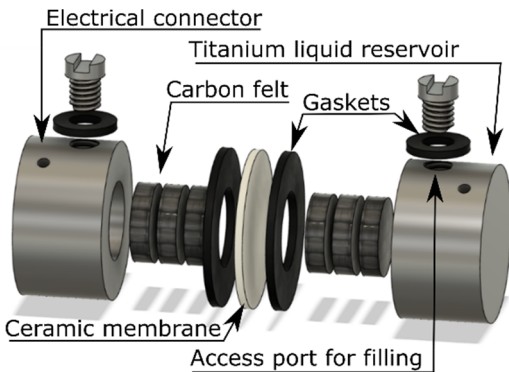

**Figure 1.** A schematic view of the main cell type used in this work.

In this work, the cells were static, i.e., no pumps or tubes were used, but titanium casing and gaskets from a variety of materials, depending on the different types of solvents, were.

The main components of redox-mediated cells are as follows: On the negative side, anode material and anolyte are combined with low voltage redox mediators such as biphenyl, benzophenone, pyrene, or any other suitable compound [23,24]. For the majority of the results in this work, however, the negative side is a more conservative configuration with sodium metal and diglyme-NaTf electrolyte.

Similarly, for the catholyte, the cathode material is combined with a high-voltage redox mediator such as sodium iodide/$I_2$. The catholyte and anolyte are any solvent-salt mixture that is stable in the required potential range. In this work, acetonitrile with 1 M sodium bis(fluorosulfonyl)imide was used as a solvent and for supporting electrolyte, respectively.

The anolyte and catholyte are kept separated by the ceramic electrolyte that should be compatible with both anolyte and catholyte while having high ionic conductivity and good mechanical properties. For this work, only commercially available β-alumina type with planar geometry was used as the membrane.

The electron transfer between the positive and negative side is enabled by the current collector that resides close to the respective sides of the ceramic membrane. In our case the current collector consisted of carbon felt in contact with the titanium casing, although nickel and aluminum mesh were also tested.

## 2. Materials and Methods

Materials used are as follows: sodium metal (99.99%, Sigma-Aldrich) was used for the metal anodes, and the anolyte consisted of 1 M sodium trifluoromethane sulfonate ($CF_3NaO_3S$, sodium triflate, Solvionics) in diglyme ($C_6H_{14}O_3$, DEGDME, Aldrich). The catholyte consisted of equimolar solutions of 0.01 M sodium iodide (>99.5%, Aldrich)/iodine

crystals (Analytical reagent, Mallinkrodt) dissolved in anhydrous acetonitrile (99.8% anhydrous, Sigma-Aldrich), with 0.5 M sodium bis(fluorosulfonyl)imide (NaFSI, Solvionic) supporting salt. A 3.18 mm piece of carbon felt (VWR 99%, 0.6 m$^2$/g) was used as the current collectors for the cathode side of the cell, while 1 mm thick sodium β″ alumina (Ionotec, Runcorn, UK) was used as the separator between the cathode and anode. Battery cycling was performed on a Landt CT2001A battery charging system™ or Neware$^®$ BTS4000 battery charging system.

Monoclinic Prussian blue used as a solid catholyte booster was obtained from Altris AB. Monoclinic Prussian blue was prepared by vacuum drying at 70 °C 12 h before use, while rhombohedral was converted from monoclinic powder by drying in vacuum at 140 °C for 15 h.

Titanium cells were constructed by machining from type 2 titanium cylinders and were designed with a 0.5 mL cavity of 10 mm inner diameter and 20 mm outer diameter for the cathode. For the anode side, a titanium well with a 0.5 mm deep cavity that holds about 50 mg of solid sodium metal was used.

The carbon felt current collector was first heated in air at 500 °C for 3 h to maximize electronic conduction, and disks were cut to size. Each liquid half-cell held 3 carbon felt disks when pressed, yielding ca. 33% compression for the carbon felt. Cells were assembled in our glovebox and placed in sealed GLS80 jars with electrical contacts to isolate the cell from ambient atmosphere as much as possible.

Clean titanium parts and any pre-made gasketing were pre-heated to at least 60 °C overnight and introduced while warm to the glovebox to minimize any surface moisture. Carbon felt and ceramic disks were heated to at least 80 °C for several hours, introduced while warm to the glovebox, and kept in sealed packages when not in use. Assembly of titanium cells was performed completely inside the glovebox, as well as any solution preparation. Sodium was pressed into the shallow well anode prior to assembly and scraped such that it is flush with titanium cell half. The catholyte cell half holds three carbon felt disks, and the Prussian blue (if used) between the carbon felt disks, away from the separator membrane. Prior to assembly, the glass separators were infused with 65 uL of diglyme/NaTf. Once assembled, the catholyte is filled using a syringe, and the cell is capped. Galvanostatic cycling between 1.7 and 3.5 V vs. Na$^+$/Nawas started with a 10 h rest period to allow wetting and an equilibrium to be established with components and progressed at 0.13 mA/cm$^2$, which corresponds to 100 µA current.

## 3. Results and Discussion

In this work, the booster consisted of high-quality and low-defect Prussian blue, both in the hydrated monoclinic form and the anhydrous rhombohedral form [21]. As can be seen in cycling from standard laminated half-cells in Figure 2, hydrated Prussian blue has a slightly sloping potential between 2.8–3.25 V vs. Na$^+$/Na while the anhydrous form has two well defined plateaus at about 3 and 3.3 V vs. Na$^+$/Na. In terms of specific capacity, Prussian blue obtains ca. 80 mAh g$^{-1}$ in the hydrated monoclinic form and 150–160 mAh g$^{-1}$ when completely dehydrated to rhombohedral structure.

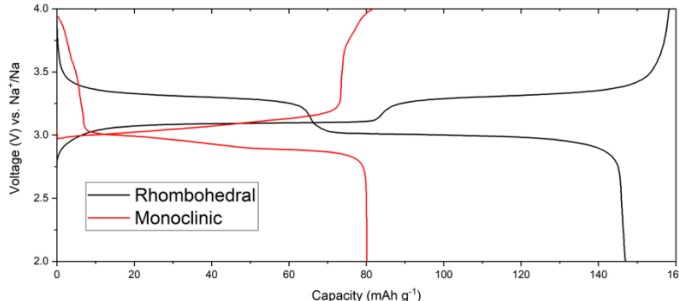

**Figure 2.** Voltage profiles of Prussian blue in half-cell pouch-cell using sodium metal anode with a hydrated monoclinic cathode in red and a dehydrated rhombohedral cathode in black.

The combination of sodium iodide and Prussian blue was chosen since sodium iodide can reduce Prussian blue while forming $NaI_3$ Equation (1), while conversely, $I_2$ formed during oxidation of $NaI_3$ will oxidize Prussian blue while regenerating $NaI_3$ Equation (2). It should be mentioned that the equations are very idealized and the oxidation of Prussian blue in particular might not be complete. The standard potentials of the $I_3^-/I$ and $I_2/I_3^-$ reactions in acetonitrile have been measured as $-0.34$ and $0.315$ V vs. $FC^+/FC$ respectively by other authors [25], and this corresponds to 2.77 V and 3.42 V vs. $Na^+/Na$ for the two reactions.

$$3NaI + Fe_2(CN)_6 \Rightarrow NaI_3 + Na_2Fe_2(CN)_6 \tag{1}$$

$$3I_2 + Na_2Fe_2(CN)_6 \Rightarrow 2NaI_3 + Fe_2(CN)_6 \tag{2}$$

The cells with 0.01 M NaI–ACN–NaFSI catholyte with and without the booster were galvanostatically cycled against Na metal. In Figure 3, the galvanostatic cycling at 0.13 mA/cm$^2$ of cells with and without different amount of monoclinic and rhombohedral Prussian blue boosters is compared. The results show that both types of boosters are reversibly cycled through the action of sodium iodide redox mediator, although how much of the capacity is attained by simple contact with the carbon felt is hard to estimate. By subtraction of the capacity obtained from the redox mediator without PB boosters, the utilized capacity of the booster can be estimated. The rhombohedral phase Prussian blue attained 97 and 111 mAh g$^{-1}$ from 10 and 20 mg booster loading, respectively, whereas monoclinic Prussian blue only reaches ca 60–66 mAh g$^{-1}$ capacity for the equivalent loadings. It seems that the higher loadings of booster actually increase utilization, although increasing the amount beyond 20 mg had the opposite effect. For the cell with 100 mg rhombohedral booster, only 61 mAh g$^{-1}$ capacity was attained, which indicates poor accessibility to the powder in the cell due to its being filled to its limit with solid powder (see Figure S1).

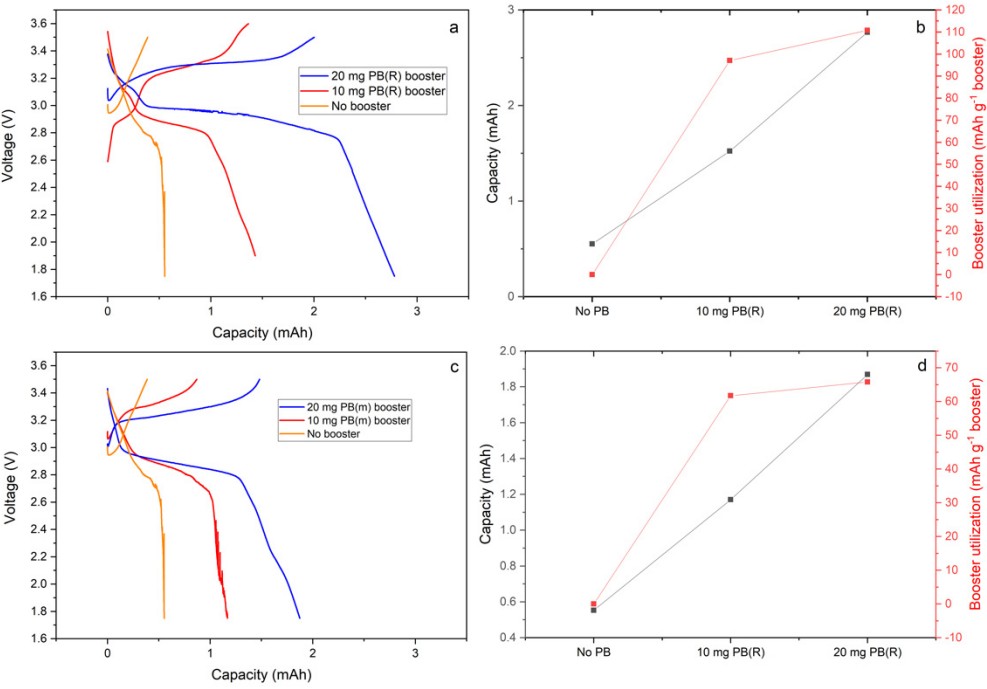

**Figure 3.** First cycle voltage profile for titanium cells with β-alumina membranes, sodium metal anodes, and 0.01 M NaI/I$_2$ redox mediator. Showing rhombohedral and monoclinic Prussian blue (PB) booster with different loadings (**a**,**c**) with the corresponding PB booster utilization and cell capacity (**b**,**d**).

The capacity of monoclinic Prussian blue at 60–66 mAh g$^{-1}$ vs. the theoretical capacity of 80 mAh g$^{-1}$ represents ca. 80% booster utilization. The missing capacity can either be explained by the conservative cut-off voltage, which might not be enough to fully utilize

the oxidation power of $I_2$, although it might also be a simple case of poor contact with the powder.

Using the same argumentation, it would be expected that ca. 80 mAh g$^{-1}$ capacity would be available for cycling from the rhombohedral phase. The sharp increase in potential from 3 to 3.3 V at 50% state of charge is expected to result in the $I_2$ being unable to oxidize the Prussian blue further than in the case for monoclinic Prussian blue. The obtained capacities are therefore surprisingly high, and no well-founded explanation for this has so far been produced. Still, 110 mAh g$^{-1}$ capacity only amounts to 68% utilization of the booster, and thus it seems that a redox mediator with more potent oxidizing abilities or a higher cut-off potential is called for to fully utilize rhombohedral Prussian blue. That being said, the potential matching between mediator and booster is complex and depends on concentration and state of charge for both mediator and booster [22].

The voltage profiles for boosted cells are quite attractive despite showing quite substantial polarization and hysteresis. The extra capacity from the boosters appears as a plateau whose length correlates well to the booster loading. The plateau appears at 3.2–3.4 V on charge and 3–2.7 V on discharge, and only a small initial extra feature of higher voltage discharge differentiates the profile of rhombohedral from that of monoclinic Prussian blue. This small feature could be attributed to the high potential plateau of rhombohedral Prussian blue and indicate that some of the booster is acting as an ordinary cathode material.

The Coulombic efficiency for the cells without booster is consistently above 100% (see Figure 4), which suggests that either the sodiation of the elemental $I_2$ initially present in the catholyte continues over several cycles, or there are unwanted reduction reactions in the catholyte during discharge. Nevertheless, the non-boosted cells retained 97% of the original capacity after 15 cycles. The energy efficiency of the cells starts out close to 100% and remains above 85% for the first 15 cycles, although severe polarization is apparent on the 15th cycle discharge (see Figure 5e).

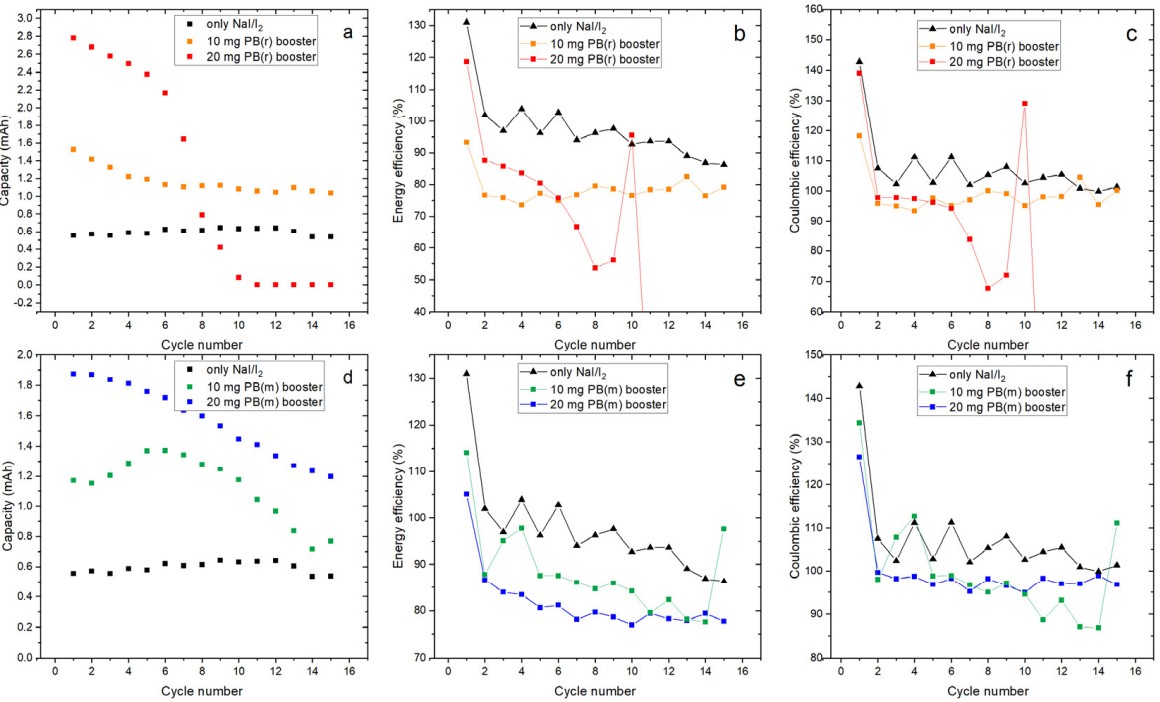

**Figure 4.** Discharge capacities, energy efficiencies and Coulombic efficiencies for the cells over the first 15 cycles using rhombohedral (**a**–**c**) or monoclinic boosters (**d**–**f**).

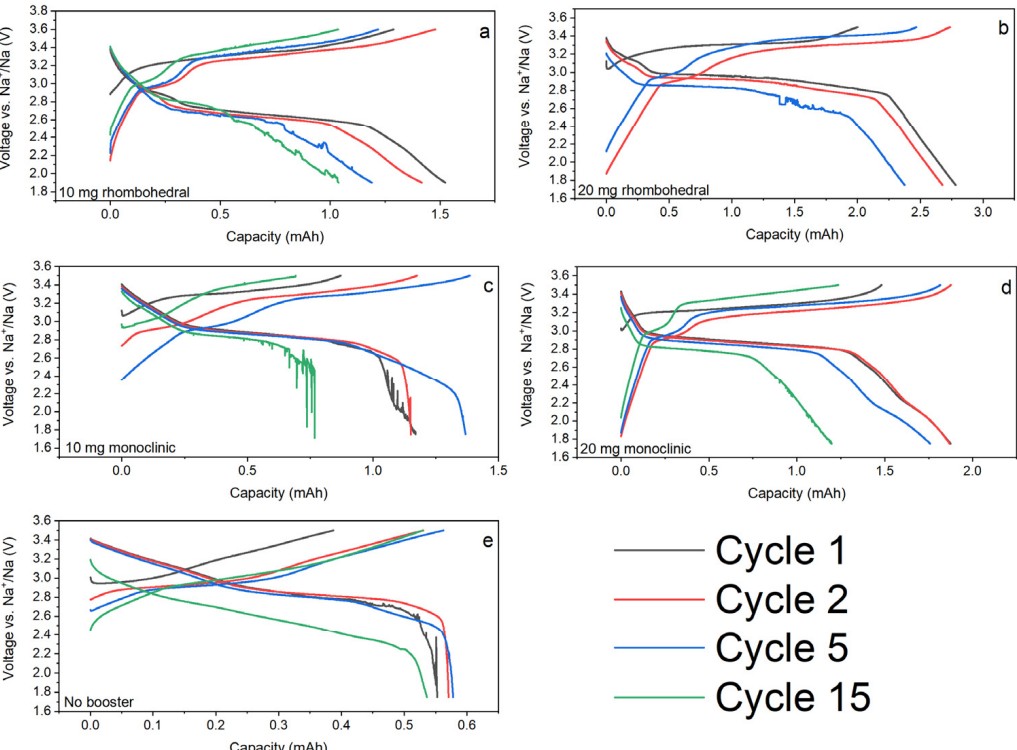

**Figure 5.** Evolution of the voltage profiles in cycles 1, 2, 5, and 15 for the cells. Booster loading was 10 mg and 20 mg rhombohedral PB (**a**,**b**) or equivalent amount of monoclinic PB booster (**c**,**d**). An identical cell without booster is also shown (**e**).

Although the utilization of solid booster and the voltage efficiency is quite promising, there is a severe issue with fading in the boosted cells (Figure 4). The boosted cells display somewhat erratic Coulombic efficiencies that start out high but decrease rapidly as polarization and subsequent capacity loss occurs. The energy efficiencies of both the monoclinic and rhombohedral boosters are quite encouraging with initial cycles reaching close to 90% and several systems remaining close to 80% at cycle 15. Although the energy efficiency is lower than laminated cells, and drops at an unacceptable rate, the high values for rhombohedral Prussian blue show that a sodium iodide redox mediator performs within acceptable parameters even for the case where two discharge plateaus exists.

Since the metallic sodium provides a generous reservoir, and the Prussian blue powders show good cycling stability in laminated cells, we assume that the rapid decrease of capacity is due to increased polarization. From the voltage profiles, it appears that the high fading ratio of the boosted cells is due to the polarization of the cell pushing the plateau for sodium extraction from PB out of the cycling window (see Figure 5). The shape of the discharge and charge profiles remain rather unchanged except for the shift that is due to increased resistance. The cause of the resistance increase can be attributed to many different parameters such as leaking cells consuming sodium and forming insoluble surface layers. One main suspect is the anode since plating and stripping experiments using both pouch cells and titanium cells showed that titanium cells have quite bad long-term performance (see Supplementary Materials Figure S2).

## 4. Conclusions

Prussian blue boosters can be used in conjunction with redox mediators. NaI enabled a capacity up to 110 mAh g$^{-1}$ from rhombohedral Prussian blue. The energy efficiency of the tested system is quite encouraging with 90% being demonstrated during the initial cycles. This work does however show that NaI/I$_2$ redox mediator might not be oxidizing enough to fully utilize the capacity of Prussian blue and other redox-mediators should

be investigated. Furthermore, the causes of the rapid capacity fading needs to be further studied as for now it can only be attributed to the polarization increase that probably stems from the poor plating and stripping in the titanium cell setup. The proposed sodium-based redox mediated cell based on non-aqueous electrolytes with relatively high-potential is an appealing system to develop for low-cost and sustainable stationary energy storage.

**Supplementary Materials:** The following are available online at https://www.mdpi.com/article/10.3390/en14227498/s1. Figure S1. First discharge voltage profile for rhombohedral and monoclinic Prussian blue a with the corresponding PB booster utilization and cell capacity b for 0, 10, 20, and 100 mg rhombohedral booster, Figure S2. Comparison on symmetrical cells with stripping and plating of 1 mAh sodium in DEGDME-NaTf electrolyte. Polarization for titanium cell setup (a) and pouch cell (b). Both cells used sodium β´´alumina as separator.

**Author Contributions:** Conceptualization, R.M.; Formal analysis and experiments, J.O.; writing—original draft preparation, R.M.; writing—review and editing and funding acquisition, R.Y. All authors have read and agreed to the published version of the manuscript.

**Funding:** The project was financially supported by the Swedish Energy Agency via project no. 47418-1 and via STandUP for Energy.

**Conflicts of Interest:** Two of the authors, Reza Younesi and Ronnie Mogensen are co-founders of Altris AB, from which the booster material was obtained.

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
