# Peer review of "High Voltage Redox-Meditated Flow Batteries with Prussian Blue Solid Booster"

_energies, doi:10.3390/en14227498_

Round 1
Reviewer 1 Report
The authors show a very nice example of solid booster flow battery, with Prussian Blue coupled iodide mediator. The work develops good experimental set-up to study these systems, although capacity fading seems to be an issue. Really nice to see boosters in non-aqueous batteries, and the energy storage density of 900 Wh/l sounds quite promising. I think this work should be accepted after minor revisions:
I would have liked an explaining section in the introduction or beginning of results where authors explain that in many redox-targeting systems, a near-perfect match in reduction potentials between booster and mediator are needed, but in the case of iodine, a larger utilization can be accessed because different oxidation states are used for the charging and discharging respectively. But this comes at the cost of the potential gap between NaI and Fe2CN6 and I2 and Na2Fe2CN6. This has been described for example in a recent review: https://doi.org/10.3390/molecules26082111
Thermodynamic analysis of the accessible SoC range would also have been welcomed.
>100% coulombic efficiency observed with NaI electrolyte only is strange. Could this be related to TiO2 layer on the Ti metal? Experiments without NaI and Prussian blue could maybe tell something about this. Also, now it seems that PB is in contact with the carbon felt, so maybe some of the reaction takes place by direct electron transfer between carbon felt and PB instead of the mediated reaction.
Some minor comments:
Line 39: ''redox-mediated system'' instead of redox mediators helps to better understand the concept.
Line 39 and 40: With the current word arrangement, it seems that instead of the fluid, redox mediators shuttle the ions, while redox mediators are ions inherently and the ions for intercalation exist in the electrolyte (fluid) and are not shuttled by the redox mediators.
Line 41: on should be one
Line 60: either ''a'' should be deleted or ''modules'' should be altered to ''module''.
Line 74: ‘‘provide’’ seems to be correct instead of ‘‘provides’’ as it refers to PB and its analogues.
Line 80: prevent is correct not prevents
Line 93 and 94: Better to be modified to: In this work the cells were static, i.e. no pumps or tubes, and used titanium casing and gaskets from a variety of materials depending on the different types of solvents.
Figure 1.A: It improves the understanding if names of different parts such as current collectors, ceramic membrane, titanium casing is brought in the figure.
Line 111: ‘‘that positive …’’ should be ’’the positive’’.
Line 130: Machining is correct
Line 139 and consequent lines: It is better to use a consistent grammatical tense throughout the article: for example using ''were'' pre-heated instead of ‘’are’’.
Line 197: 50 ‘‘%’’ state of charge
Line 216: Better to already mention that in here Figure 4 is being discussed. Figure 4 is not referred to until line 220.
12: Though instead of Thought
49: one instead of on
93: were used, and…
102: positive the catholyte?
130: constructed by machined?
145: Metal cell half?
192: might NOT be?
223: utilization OF solid booster
227: ARE quite encouraging
Author Response
We thank the reviewer, we have put the answers to the comments into the attached word file

Reviewer 2 Report
In this manuscript, Prussian blue solid boosters for use in high voltage redox-mediated flow batteries (RMFB) based on non-aqueous electrolytes are reported. The system consisted of NaI as a redox mediator in an acetonitrile catholyte containing solid Prussian blue powder. NaI enabled a capacity up to 110 mAh g-1 from rhombohedral Prussian blue. The proposed sodium-based RMFB is an appealing system to develop for low-cost and sustainable stationary energy storage.
I consider the content of this manuscript will definitely meet the reading interests of the readers of the Energies journal. Generally speaking, English writing is very good. However, I suggest the author check the full text for possible grammar errors. I will point out some of them, but cannot cover all of them.
Therefore, I suggest giving a minor revision and the authors need to clarify some issues or supply some more data to enrich the content.
My detailed comments are attached to a PDF file.

Author Response
We thank the reviewer for the review, we have collected all the answers and actions done based on the comments i the the attached word file.
